# Antiproliferative and Antimicrobial Potentials of a Lectin from *Aplysia kurodai* (Sea Hare) Eggs

**DOI:** 10.3390/md19070394

**Published:** 2021-07-14

**Authors:** Rubaiya Rafique Swarna, A. K. M. Asaduzzaman, Syed Rashel Kabir, Nawshin Arfin, Sarkar M. A. Kawsar, Sultana Rajia, Yuki Fujii, Yukiko Ogawa, Keisuke Hirashima, Nanae Kobayashi, Masao Yamada, Yasuhiro Ozeki, Imtiaj Hasan

**Affiliations:** 1Department of Biochemistry and Molecular Biology, University of Rajshahi, Rajshahi 6205, Bangladesh; rubaiyarafique5252@gmail.com (R.R.S.); jonyasad2005@yahoo.com (A.K.M.A.); rashelkabir@ru.ac.bd (S.R.K.); arfin.nawshin@gmail.com (N.A.); 2Department of Chemistry, University of Chittagong, Chittagong 4331, Bangladesh; akawsarabe@yahoo.com; 3Center for Interdisciplinary Research, Varendra University, Rajshahi 6204, Bangladesh; rajia_bio@yahoo.com; 4School of Science, Yokohama City University, 22-2 Seto, Kanazawa-ku, Yokohama 236-0027, Japan; n195274c@yokohama-cu.ac.jp (K.H.); kobayashi.nanae444@gmail.com (N.K.); yamada.mas.ug@yokohama-cu.ac.jp (M.Y.); ozeki@yokohama-cu.ac.jp (Y.O.); 5Graduate School of Pharmaceutical Sciences, Nagasaki International University, 2825-7, Huis Ten Bosch-cho, Sasebo, Nagasaki 859-3298, Japan; yfujii@niu.ac.jp (Y.F.); yogawa@niu.ac.jp (Y.O.); 6School of Medicine, Yokohama City University, 3-9, Fukuura, Kanazawa-ku, Yokohama 236-0004, Japan

**Keywords:** antibacterial activity, anticancer activity, antifungal activity, *Aplysia kurodai*, apoptosis, Ehrlich ascites carcinoma, lectin

## Abstract

In recent years, there has been considerable interest in lectins from marine invertebrates. In this study, the biological activities of a lectin protein isolated from the eggs of Sea hare (*Aplysia kurodai*) were evaluated. The 40 kDa *Aplysia kurodai* egg lectin (or AKL-40) binds to D-galacturonic acid and D-galactose sugars similar to previously purified isotypes with various molecular weights (32/30 and 16 kDa). The N-terminal sequence of AKL-40 was similar to other sea hare egg lectins. The lectin was shown to be moderately toxic to brine shrimp nauplii, with an LC_50_ value of 63.63 µg/mL. It agglutinated Ehrlich ascites carcinoma cells and reduced their growth, up to 58.3% in vivo when injected into Swiss albino mice at a rate of 2 mg/kg/day. The morphology of these cells apparently changed due to AKL-40, while the expression of apoptosis-related genes (p53, Bax, and Bcl-XL) suggested a possible apoptotic pathway of cell death. AKL-40 also inhibited the growth of human erythroleukemia cells, probably via activating the MAPK/ERK pathway, but did not affect human B-lymphoma cells (Raji) or rat basophilic leukemia cells (RBL-1). In vitro, lectin suppressed the growth of Ehrlich ascites carcinoma and U937 cells by 37.9% and 31.8%, respectively. Along with strong antifungal activity against *Talaromyces verruculosus*, AKL showed antibacterial activity against *Staphylococcus aureus*, *Shigella sonnei*, and *Bacillus cereus* whereas the growth of *Escherichia coli* was not affected by the lectin. This study explores the antiproliferative and antimicrobial potentials of AKL as well as its involvement in embryo defense of sea hare.

## 1. Introduction

Lectins are carbohydrate-binding proteins, omnipresent in almost all life forms, which specially recognize carbohydrate structures of glycoproteins and glycolipids present on the cell surface [1]. Marine invertebrates possess diverse classes of lectins with various protein foldings and different carbohydrate-binding specificities [1,2,3,4,5]. Many of these lectins show affinity towards galactose-related carbohydrates and perform biological functions including antitumor and antimicrobial activities.

Endogenous lectins interact with cell-surface glycans to mediate numerous biological functions in cells [6]. Compared to normal cells, glycosylation pattern of cancer cells are altered, which affects multiple cellular mechanisms performed by lectins through their association with corresponding glycans [7]. As a result, this alteration supports their neoplastic progression [8,9]. For example, galactose-binding lectins or galectins aid the binding of tumor cells through the interaction with galactose-containing carbohydrate ligands on tumor cells [9]. Many marine invertebrate lectins have been reported to bind with glycans on the tumor cell surface and killed those inducing apoptosis [3,10,11,12]. On the other hand, lectins can also modulate the entry and subcellular targeting of drugs into cancer cells [13,14]. Both these phenomena exhibit considerable influence of lectin–glycan interaction on the proliferation and regulation of tumor cells [5,10,11,12,14].

As a component of innate immunity, lectins present in marine invertebrates agglutinate both Gram-negative and Gram-positive bacteria through the interaction with lipopolysaccharides and peptidoglycans on their cell walls [4,5,15,16]. Galactose residues present as terminal sugars in lipopolysaccharides (LPS) of bacteria impact the intracellular composition of bacteria and maintain the synthesis of UDP-galactose for LPS [17,18]. Bacterial endotoxins also possess galactose as a constituent [19]. On the contrary, antifungal activity of lectins is not well reported yet.

Sea hares are marine gastropod mollusks found in common coastal areas. Various lectins have been reported to be present in eggs and gonads of sea hares due to their possible roles in early development and body defense. D-galacturonic acid and D-galactose-binding lectins with molecular masses 16–34 kDa have been obtained from the eggs of *Aplysia kurodai* [11,20,21,22] and other *Aplysia* species (*A. depilans*, *A. dactylomela,* and *A. californica*) [23,24,25]. Primary structures of all these previously purified egg lectins possess a novel triple tandem repeating sequence consisting of 210–230 amino acids and show striking similarities to domain DUF3011 of some uncharacterized bacterial proteins [26].

These Aplysia egg lectins showed cytotoxic activities against certain tumor cells and were involved with organogenesis in early developmental stages. They might have played protective roles in marine organisms as well. Hasan et al. reported an α-galactose-binding 40 kDa lectin from eggs of AKL in 2014 that inhibited streptolysin O-induced hemolysis and growth of *Streptococcus pyogenes* [22].

The 40 kDa lectin from *Aplysia kurodai* eggs will be designated here as AKL-40, to differentiate it from other AKLs. We characterized this egg lectin and determined its N-terminal sequence. Previously, in vitro antiproliferative activity of two AKLs was reported [11,21]. In this study, antiproliferative activities of this particular lectin were determined for the first time in vivo, using Swiss albino mice. In addition, other biological properties, such as in vitro anticancer activity against different cancer cell lines and antibacterial potential of the lectin, was further evaluated along with its antifungal activity.

## 2. Results

### 2.1. Purification, Confirmation of the Molecular Mass and Hemagglutination Activity of Aplysia kurodai Egg Lectin (AKL-40)

Affinity chromatography showed different molecular masses of AKLs found in the eggs of *Aplysia kurodai* (Figure 1). After applying the crude protein sample, the melibosyl-agarose affinity chromatography column was washed with TBS and bound lectins were eluted with 10 mM D-galacturonic acid or D-galactose-containing TBS. Each polypeptide was separated with gel filtration chromatography by using Sephacryl S-200HR, (Cytiva, Marlborough, MA, USA) (data not shown). In the reducing condition, polypeptides with molecular weights of 40 kDa, 32/30 kDa, and 16 kDa were denoted as AKL-40, AKL, and AKL-2, respectively, in this study (Figure 1: white triangle: AKL-40; black triangles: AKL and AKL-2). The minimum concentration of AKL-40 to agglutinate human and mice erythrocytes was found to be 24 and 32 µg/mL, respectively.

### 2.2. N-Terminal Amino Acid Sequence of AKL-40

The N-terminal region of AKL-40, including the first 35 amino acids of the polypeptide, was determined by Edman degradation with a repetitive yield of 84.19% (Appendix A). This partial N-terminal sequence of AKL-40 fitted with those of AKL-a to -d and ADEL, showing similarities (Figure 2). Unlike others, AKL-40 possessed a forward extension comprising four additional amino acids.

### 2.3. Toxicity of AKL-40 against Brine Shrimp Artemia nauplii

A dose-dependent graph showed that 70% of shrimp nauplii died at a concentration of 160 µg/mL of AKL-40 (Figure 3). The LC_50_ value was determined to be 63.63 µg/mL indicating AKL-40 as a moderately toxic protein.

### 2.4. In Vivo Anticancer Activity of AKL-40 against Ehrlich Ascites Carcinoma Cells, Their Morphological Changes, and Expression of Apoptosis-Related Genes

After administering 1 mg/kg/day and 2 mg/kg/day of AKL-40 for five days, growth of EAC-cells in the lectin-treated mice (Groups B and C) was inhibited to 28.7% and 58.3%, compared to the untreated (or control) mice from group A (Figure 4A). Weight of the treated mice reduced significantly compared to the untreated mice (data not shown). EAC cells from untreated mice were spherical and regular-sized (Figure 4B, column 0). Irregular-shaped kidney-bean-like EAC cells from the treated mice groups (Figure 4B, column 1 and 2). Upregulation of p53 gene expression was observed in EAC cells from group C, though it was a weak one. Expression of Bax gene was there in EAC cells from lectin-treated mice whereas no expression was found in EAC cells from untreated mice. Cells from AKL-40-treated mice showed no expression of the Bcl-X_L_ gene whereas, in EAC cells from control mice, it was visible. Expression of GAPDH gene in EAC cells from both untreated and lectin-treated mice confirmed the quality of mRNA isolated (Figure 4C).

### 2.5. In Vitro Anticancer Activity of AKL-40 against Different Cancer Cells and Activation of Signal Transduction Molecules

AKL-40 significantly inhibited the growth of K562 cells at a concentration of 100 µg/mL or more (Figure 5A, orange bar). On the other hand, the lectin could not influence the growth of human B-lymphoma cells (Raji) and rat basophilic leukemia cells (RBL-1) (Figure 5A, blue and gray bars). During a period of 24 h, extracellular signal-regulated kinase (ERK)_1/2_ and p38kinase molecules became phosphorylated in AKL-40 treated K562 cells, as shown by western blotting (Figure 5B).

Minimum agglutination concentrations of AKL-40 for EAC and U937 cells were 20 and 24 µg/mL, respectively. The dose-dependent effect of AKL-40 against U937 and EAC cells was observed by MTT assay. At lower concentrations (100 µg/mL), growth inhibition of both cell types was similar, but at higher concentrations, AKL-40 showed slightly higher activity against EAC cells. At the concentration of 250 µg/mL, the percentage of growth inhibition for EAC cells was 37.9%, compared to 31.8% for U937 cells (Figure 5C).

### 2.6. Antimicrobial Activity of AKL-40

AKL-40 showed antibacterial activity against *Staphylococcus aureus*, *Bacillus cereus,* and *Shigella sonnei*. No zone of inhibition was there for *Escherichia coli*. Slightly larger zones were observed around the disks soaked with higher doses (100 µg/disc) of AKL-40 in case of every susceptible bacterial species (Figure 6A). Maximum activity of AKL-40 was found against *Staphylococcus aureus* in terms of the inhibition zones formed. Growth of *Talaromyces verruculosus* was also very efficiently inhibited by AKL-40. The fungi rapidly grew and after two weeks, nearly covered the whole petri dish. However, when discs soaked with AKL-40 were placed in the fungal media, almost no growth of the mold was observed (Figure 6B).

## 3. Discussion

Following the purification of two lectins AKL and AKL-2 from *Aplysia kurodai* eggs, a third lectin with different molecular mass (40 kDa) had been purified [11,21,22]. This study focused on the biological activities of AKL-40 and the comparison of those activities to other marine lectins, specially isolated from mollusks. A number of lectin families have been reported in the phylum Mollusca whereas the Aplysia egg lectin family is structurally unique consisting of various isotypes. Galacturonic acid has been found in polysaccharides present in mollusks, such as cuttlefish [27]. Eggs of *Aplysia kurodai* also contained galactose/galacturonic acid-binding lectins as a mixture of 40, 32/30, and 16 kDa polypeptides (Figure 1). Such diversity of galactose/galacturonic acid-binding lectins in Aplysia eggs perhaps is a result of their response to changing marine environments. It might be interesting to find out the function of each molecular species of Aplysia egg lectins during each developmental stage of the embryo. The primary structure of the sea hare egg lectin family is different from other lectins because of the presence of homolog sequences that are also found in organisms, such as bacteria and brachiopods [26]. A partial but novel N-terminal sequence of AKL-40, consisting of 35 amino acids, indicated that this polypeptide was different from other variants (such as AKL-a to -d) by having a four-amino acid forward extension (Figure 2). This result also showed low sequence similarity of AKL-40 to ADEL.

Determining the level of toxicity of lectins could be important to study and predict their structures, physiological functions, and biological applications. With an LC_50_ value of 63.63, AKL-40 was moderately toxic to brine shrimp nauplii (Figure 3) whereas AKL-2, the lactose-binding counterpart, was more than three times toxic [28]. However, toxicity of lectins is not always related to cell regulatory effects as their interactions with glycans play vital roles in cell signaling. Despite having much lower LC_50_ value (384.53 µg/mL), MytiLec-1, a lectin from another marine mussel, could kill Burkitt lymphoma and U937 cells in vitro, as well as Ehrlich ascites carcinoma cells in vivo [3,29]. A variable level of toxicity with different LC_50_ values (850.1, 142.1, 9.5, and 6.4 μg/mL) was also observed in lectins purified from marine sponges and a sea cucumber [30,31].

Ehrlich ascites carcinoma cells originated from mammary tissues, are spontaneous, differentiated, transplantable, and aggressive in nature. These cells grow in certain mice strains and are approved worldwide as a standard mice model [32]. AKL-40 agglutinated EAC cells via galactosyl ligands present in their cell membrane [33,34]. In a previous report, another Molluscan D-Gal-binding lectin, MytiLec-1 strongly agglutinated EAC cells at a minimum concentration of 16 µg/mL and inhibited 28 and 49% of their growth at doses of 1 and 2 mg/kg/day, respectively [29]. In the present study, minimum agglutination concentration of AKL-40 for EAC cells was 20 µg/mL and, compared to MytiLec-1, similar growth inhibition activities (28.70% and 58.32% at doses of 1 and 2 mg/kg/day) have been observed (Figure 4A). Lectin-treated cancer cells presented Irregular shapes and nuclear condensation compared to the control (or untreated) cells (Figure 4B).

Appearance of a faint band of p53 indicated to the transcription of a pro-apoptotic gene like Bax in AKL-40 treated Ehrlich ascites carcinoma cells. Expression of Bcl-X_L_, an anti-apoptotic gene had also become downregulated in the treated cells (Figure 4C). Similar expressions of p53, Bax, and Bcl-X_L_ genes have been found when MytiLec-1 was applied on EAC cells [29]. A previous study reported a lactose-binding lectin from the marine sponge *Cinachyrella apion* to induce the apoptotic death of HeLa cells through activation of Bax protein. The anti-apoptotic Bcl-2 protein showed no significant change in expression, compared to the control [35]. Evaluation of gene expression of MCF-7 cells revealed that marine red alga *Solieria filiformis* triggered caspase-dependent apoptosis. The anti-apoptotic gene Bcl-2 became down expressed, whereas the pro-apoptotic Bax gene underwent over-expression [36]. Downregulation of the anti-apoptotic factor Bcl-2 was determined also in the case of two other lectins from Sea bass (*Dicentrarchus labrax*) and sea urchin (*Strongylocentrotus purpuratus*) [37]. Therefore, it can be suggested that AKL-40 possibly calling the Bcl-2 apoptotic protein family into play marks its ability to promote apoptosis.

Marine invertebrate lectins have already been reported to exhibit antiproliferative properties, promote apoptosis, and block angiogenesis. In contrast, tumor-derived galactose-binding lectins, especially galectins, could compromise the anti-tumor immune response of CD8+ T cells to escape from the host immune surveillance [38,39]. AKL-40 significantly reduced the growth of erythroleukemia K562 cells compared to Raji and RBL-1 cells (Figure 5A). Phosphorylation of two major mitogen-activated protein kinases, p38 and ERK 1/2 became significantly upregulated by the administration of AKL-40, whereas no phosphorylation was observed for and c-Jun N-terminal kinase (JNK) (Figure 5B). It can be suggested that like other lectins previously reported, the lectin might have participated in a signaling cascade controlling cellular responses, probably leading to apoptosis [3,5,14]. In another in vitro study, AKL-40 inhibited the growth of two other cell lines, EAC and U937 (Figure 5B). U937 is a monocytic human myeloid leukemia cell line frequently used in biomedical research [40]. Compared to AKL-40, MytiLec-1 and Ricin inhibited growth of these cells in vitro at much lower concentrations, which also indicates the difference of glycan recognition for these three lectins [30,41,42].

In marine organisms, lectins recognize and bind to the surface polysaccharides of a variety of bacteria. They have ability to kill bacteria or inhibit their growth and thereby contribute to defense against infection as a part of the innate immune system [43,44]. Similar to AKL, AKL-40 also exhibited growth inhibitory activity against both Gram-positive and Gram-negative organisms (Figure 6A) [45]. *Staphylococcus aureus* were the most susceptible bacteria to both lectins. Unlike AKL, another egg lectin from *Aplysia dactylomela* (ADEL) agglutinated *Staphylococcus aureus* cells, but could not inhibit their growth [24]. A mannose/galactose-binding C-type lectin isolated from bay scallop *Argopectenirradians* also displayed this property against Gram-positive *Staphylococcus aureus* and Gram-negative *Escherichia coli* and *Vibrio anguillarum* [46]. However, due to unknown reasons, AKL, AKL-40, and ADEL could not affect the growth of *Escherichia coli* [24]. On the other hand, CvL from marine sponge *Cliona varians* showed cytotoxic effect on Gram-positive bacteria, such as *Bacillus subtilis* and *Staphylococcus aureus,* but could not inhibit the growth of Gram-negative bacteria like *Escherichia coli* and *Pseudomonas aeruginosa* [15]. These results suggest that the bactericidal activity of lectins depend not only on glycan-binding specificities, but also on multiple systems, such as multivalency or binding constancy.

Antifungal properties are not very common in lectins isolated from marine invertebrates [47]. *Aplysia depilans* gonad lectin was previously used to study the distribution of galacturonic acids in the cell walls of some pathogenic fungi [48]. In this study, high concentration (400 µg/mL) of AKL-40 totally inhibited the growth of *Talaromyces verruculosus* (Figure 6B). A study by Ruperez et al. reported the presence of galactose sugars in the cell wall of *Talaromyces verruculosus,* which justified our findings [49]. Another antifungal protein (Aplysianin E) from *Aplysia kurodai* eggs completely suppressed the growth of *Saccharomyces cerevisiae* and *Candida albicans* at a concentration of 16 µg/mL [50]. AKL also repressed the mycelial growth of *Curvularia lunata* at 100 µg/mL whereas growth inhibitory effects of two other lectins from Japanese black sponge (*Halichondria okadai*) and mussel (*Crenomytilus grayanus*) were found against *Aspergillus niger* and *Pichia pastoris*, respectively [5,45,51]. It can be postulated that certain polysaccharides present in fungal cell walls interacted with these lectins and inhibited their growth by disturbing spore germination, growth of mycelium, and synthesis of chitin, to alter the fungal cell wall [52]. Like eggs from other organisms, sea hare eggs are found lying exposed in the seashore, vulnerable to the attack of predators and microbes [53]. Therefore, egg lectins might have a role to serve as protective molecules.

## 4. Materials and Methods

### 4.1. Preparation of the Crude Extract

The eggs of sea hare were gathered from the Zushi coast, Kanagawa, Japan. Eggs were crushed in a mortar, solubilized with Tris-buffered saline or TBS (10 mM Tris(hydroxymethyl)aminomethane-HCl, pH 7.4, with 150 mM NaCl). Then, crushed eggs were blended with 10 volumes of TBS containing a protease inhibitor (10 mM, Protease Inhibitor Cocktail 100X, Wako Pure Chemical Corp, Osaka, Japan).The homogenized sample was filled up in 500-mL centrifuge bottles and centrifuged at 14,720× *g* for 1 h at 4 °C. A Suprema 21 centrifuge equipped with an NA-18HS rotor (TOMY Co. Ltd., Tokyo, Japan) was used for this step. After repeated centrifugation of the supernatant at the same speed for the same duration, a clear solution was prepared and stored as the crude extract.

### 4.2. Purification of the Lectin

The lectin was purified according to a previously described procedure [22]. The crude extract was twice centrifuged at 27,500× *g* for 1 h at 4°C and was administered to a 5 mL melibiose-agarose affinity column (J-Oil Mills Inc., Tokyo, Japan). This column was connected to a 5 mL Sephadex G-75 pre-column. After loading on the crude protein sample, the column was washed well with TBS. Lectins bound to the column were eluted with 10 mM D-galacturonic acid or D-galactose-containing TBS. The mixture of AKLs was separated by using a gel filtration chromatography of Sephacryl S-200 (Cytiva, Marlborough, MA, USA) connected to a fraction collector (FRC-10A, Shimadzu Corporation, Tokyo, Japan). Molecular weights of the lectins were confirmed by SDS-PAGE using standard marker proteins [54]. The 40 kDa polypeptide species was denoted as ‘AKL-40′ in this study.

### 4.3. Determination of N-Terminal Partial Amino Acid Sequence of AKL-40

The N-terminal sequence of the 35 amino acids of AKL-40 was determined with automated Edman degradation by using a protein/peptide sequencer PPSQ (Shimadzu Corporation, Kyoto, Japan) [55].

### 4.4. Determination of the Toxicity of AKL-40 by Brine Shrimp Nauplii Lethality Assay

The assay was performed according to a method reported earlier [29]; 4 mL of artificial seawater was taken in test tubes. Ten brine shrimp nauplii were taken in each vial and AKL-40 was added to these vials to adjust its concentrations from 10 to 160 µg/mL. There were control vials containing only seawater and nauplii, but no lectin. The experiment was repeated thrice at 30 °C for 24 h with 6 h of light exposure. Percentages of mortality of the nauplii were determined for each concentration and the LC_50_ value of AKL-40 was also determined according to the method of Finney [56].

### 4.5. In Vivo Anticancer Activity of AKL-40 against Ehrlich Ascites Carcinoma Cells Grown in Swiss Albino Mice

The in vivo experiment was approved by the Institutional Animal, Medical Ethics, Bio-safety and Bio-security Committee (IAMEBBC) for experimentations on animals, human, microbes, and living natural sources, Institute of Biological Sciences (IBSc), University of Rajshahi, Bangladesh (memo no. 102(6)/320-IAMEBBC/IBSc). Four to six weeks old Swiss albino mice (weight range 25–30 g) of both genders were collected from ICDDR’B (International Center for Diarrheal Diseases Research, Bangladesh) and EAC cells were propagated into these mice by a bi-weekly intraperitoneal transformation. Cells in ascitic fluid were drawn from a donor mouse bearing 6–7 days old tumor cells and with the help of a hemocytometer, adjusted to 2 × 10^6^ cells/mL. Normal saline was used for the dilution. Viability of tumor cells was checked by 0.4% trypan blue assay.

The mice were randomly distributed into three groups, consisting of six mice in each group. These groups were denoted as ‘A’ or control, ‘B’ or lectin-treated with lower dose (1 mg/kg/day)’ and ‘C’ or lectin-treated with higher dose (2 mg/kg/day). Moreover, 0.1 mL of cellular suspension containing viable EAC cells was injected intraperitoneally to each Swiss albino mouse. After 24 h, both lectin-treated groups (B and C) were treated for five days with an intraperitoneal injection of AKL-40 at doses of 1.0 and 2.0 mg/kg/day. All mice in groups A, B, and C were weighed to check the rate of tumor growth. Mice were sacrificed on the sixth day and EAC cells were collected from the ascitic fluid. The total number of viable EAC cells in every mouse of the treated groups (B and C) was compared to those of the control group (A) using the following formula:Percentage of inhibition = 100 − {(cells from AKL-40 treated mice/cells from control mice) × 100}

### 4.6. Morphological Observation of AKL-40 Treated EAC Cells by Fluorescence Microscope

Morphological changes of control (or untreated) and AKL-40 treated EAC cells were observed using fluorescence microscopy (Olympus iX71, Seoul, Korea). EAC cells from mice treated with and without AKL-40 for five consecutive days were collected. After washing with phosphate buffer saline (PBS), the cells were stained with 0.1 µg/mL of Hoechst-33342 at 37 °C for 20 min in dark, washed again with PBS, and observed in the microscope.

### 4.7. RNA Isolation and Checking the Expression of Apoptosis-Related Genes from Ehrlich Ascites Carcinoma Cells

EAC cells from mice treated with and without AKL-40 were collected and the total RNA was isolated using a reagent kit (Tiangen Biotech Co., Beijing, China). Concentration and purity of the isolated RNA were checked at 260 and 280 nm using a spectrophotometer. cDNA samples were prepared following the protocol of Applied Biosystems, Waltham, MA, USA, and bands for all PCR reactions were visualized in 1.4% agarose gel with a gel documentation system (Cleaver Scientific Ltd., Rugby, UK). Ten µg/mL of ethidium bromide solution was used to stain the gel. GAPDH was used as a housekeeping gene to compare with the standard. GeneRuler 1000 bp DNA ladder (Fermentas, Waltham, MA, USA) was used as marker. Specific oligonucleotides (Integrated DNA Technologies or IDT, Singapore) like p53, Bax, Bcl-X_L_ and GAPDH generated 458 bp, 477 bp, 780 bp, and 475 bp amplification products, respectively. Primer sequences of these genes under study are provided in Table 1**.**

For gene amplification, a program was set in a thermal cycler (Gene, Atlas 482, Tokyo, Japan) at 95 °C for 3 min, followed by 35 cycles of 95 °C for 30 s, 55 °C for 30 s, 72 °C for 50 s, finally at 72 °C for 10 min, and then was eventually held at 20 °C. In case of Bax and Bcl-XL, the annealing temperature was 54 °C instead of 55 °C.

### 4.8. Determination of Cytotoxic Activity of AKL-40 against Cancer Cell Lines and Detection of Activated Signal Transduction Molecules

Cytotoxic activity of AKL-40 against cancer cell lines was evaluated according to a previous report [5]. Three leukemia cell lines K562, Raji, and RBL-1 (2 × 10^5^ cells) were seeded into a 96-well titer plate and treated with different concentrations of AKL-40 for 24 h at 37 °C. Ten micro liter of WST-8 solution was added to each well and incubated for 4h at the same temperature. The absorbance was measured at 450 nm to assay the reduction in proportion of living cells by a GloMax Multi Detection System (Promega, Madison, WI, USA).

K562, Raji, and RBL-1 (2 × 10^5^ cells) were cultured with AKL-40 (0–1600 μg/mL) for 24 h, and lysed with 200 μL cell lysis buffer M. The cell lysate was separated by SDS-PAGE and electroblotted onto PVDF membrane. Primary antibodies used were directed to phospho-ERK1/2 (extracellular signal-regulated kinase), phospho-P38 (P38 MAP kinase), and phospho-JNK (c-Jun N-terminal kinase) by using each mouse mAb (Becton Dickinson, Franklin Lakes, NJ, USA) at the dilution of 1/3000. Membrane was masked with TBS containing 1% BSA, soaked with 2% Triton X-100 at RT, incubated with HRP-conjugated goat anti-mouse IgG for mouse mAb (Tokyo Chemical Industry Co., Tokyo, Japan) for 1 h, and colored with EzWestBlue (ATTO Corp., Tokyo, Japan).

### 4.9. Determination of the Minimum Agglutination Concentration of EAC and U937 Cells by AKL-40

U937 cells (ATCC CRL-3253) were collected from Yokohama City University, Japan. Fifty µL of 20 mM Tris-HCl buffer saline containing 10 mM CaCl2 (pH 7.8) was taken in each well of two U-bottomed 96-well microtiter plates (one for EAC and the other for U937 cells). AKL-40 (50 µL) was added to the titer plates through serial dilution. The number of EAC and U937 cells in RPMI-1640 media was counted using a hemocytometer (Hirschmann EM Techcolor, Eberstadt, Germany) and around 5 × 10^5^ cells were seeded in each well of the two plates. The plates were agitated for 5 min in a microshaker, kept at room temperature for 30 min, and agglutination titers for both cell types were recorded.

### 4.10. Anticancer Activity of AKL-40 in Vitro against U937 and Ehrlich Ascites Carcinoma Cells

One hundred µL of RPMI-1640 media was taken in two 96-well flat bottom titer plates (one for EAC and the other for U937 cells). AKL-40 was added to the wells at final concentrations of 0, 100, 200, and 250 µg/mL. 100 µL of EAC cells (collected from mice) and U937 cells were added to each well (5 × 10^5^ cells/well). There were three control wells containing only cancer cells. After an incubation period of 24 h in 5% CO_2_ incubator at 37 °C, the clear supernatant was carefully removed from each well. Then 180 µL of PBS and 20 µL of 3-(4,5-dimethylthiazol-2-yl)-2,5-diphenyl tetrazolium bromide or MTT (5 mg/mL) were added and both plates were kept in the incubator for 8 h at 37 °C. The supernatant was removed again from each well, 200 µL of acidic isopropanol was added, and absorbance of each well was recorded by a titer plate reader at 570 nm. Percentages of cell proliferation inhibition were calculated by the following equation:Proliferation inhibition ratio (%) = (A − B) × 100/A
where A is the OD_570_ nm of the cellular homogenate from control wells and B is the OD_570_ nm of the cellular homogenate from wells treated with AKL-40.

### 4.11. Bactericidal Activity of AKL against Pathogenic Bacteria

Antibacterial activity of AKL-40 was determined by agar disc diffusion method against four bacteria—two Gram-positive (*Staphylococcus aureus* and *Bacillus cereus*) and two Gram-negative (*Shigella sonnei* and *Escherichia coli*). Petri dishes containing lysogeny broth (LB) media for each bacterial species were prepared and four paper discs (control, high dose, low dose, and antibiotic) were placed on each petri dish. Moreover, two hundred and one hundred µg of AKL-40 were used as high and low doses, respectively, whereas the antibiotic disc contained 15 µg of ampicillin. Control disc was soaked only with LB media. After keeping at 37 °C for 24 h in an incubator, zones of inhibition around the discs were observed in each petri dish.

### 4.12. Antifungal Activity of AKL-40 against Talaromyces verruculosus

*Talaromyces verruculosus*, a fungus from the division ascomycota, was cultured in petri dishes with potato dextrose agar (HiMedia Laboratories Pvt. Ltd., Mumbai, India) as a medium following the standard procedure. In the ‘Test’ petri dish, three discs soaked with 400 µg/mL of AKL-40 were placed around whereas the ‘Control’ petri dish had only one disc, soaked only with media. After keeping at 25 °C for 1 week in an incubator, growth of the fungi in these petri dishes was observed. After another week, there was a second observation to find any growth inhibition.

### 4.13. Statistical Analysis

All experiments of this research work were performed in three replicates. Experimental data were expressed as mean ± SD. To test differences between experimental conditions, one way ANOVA and Dunnett’s post-test correction was used. The results were statistically significant when ** *p* < 0.01 and * *p* < 0.05.

## 5. Conclusions

This study might be the start of isolating and sequencing amino acids in the specific peptides/proteins accountable for the observed antiproliferative effects of AKL-40. Abolition of biological activities by AKL-40 was not checked in the presence of inhibiting sugars, such as D-galacturonic acid or D-galactose, which is a limitation of this study. Determination of MIC and MBC values for the bacteria and fungi could also be performed. Additional investigations are required to investigate the association of lectin in infection and pathogenesis of different bacteria, as well as to recognize the molecular mechanism of signaling pathways activated during infection, which could help in the development of new therapeutic approaches. It is quite interesting to predict that a protein domain from bacteria was evolved in the course of time to exert antimicrobial effects in marine organisms. Such predictions could be evaluated in the future via a combined study of functional genomics, transcriptomics, and glycobiology of Aplysia lectins.

## Figures and Tables

**Figure 1 marinedrugs-19-00394-f001:**
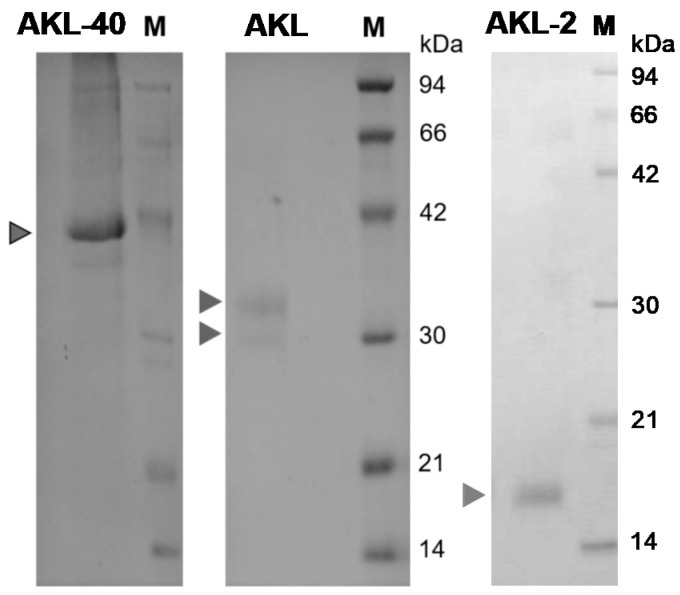
Different molecular masses of AKLs found in the eggs of *Aplysia kurodai.* AKL mixtures obtained from the melibiose column were combined and separated by using Sephacryl S-200 gel filtration column (black triangles show the polypeptides indicating AKLs). AKL-40: 40 kDa species (used in this study), AKL: 32/30 kDa species, and AKL-2: 16 kDa species. M: standard markers of Phosphorylase b (94 kDa), serum albumin (66 kDa), ovalbumin (42 kDa), carbonic anhydrase (30 kDa), trypsin inhibitor (21 kDa), and lysozyme (14 kDa). All samples are separated in the reducing condition and 15% polyacrylamide gels are used except AKL-2, which was separated by a 12% gel.

**Figure 2 marinedrugs-19-00394-f002:**
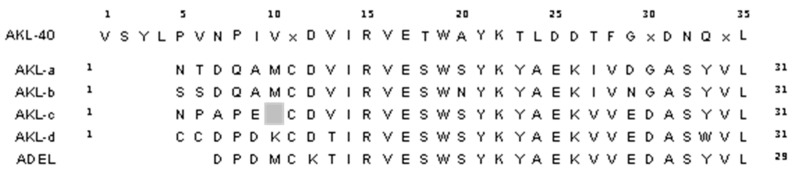
Comparison of N-terminal amino acid sequences of AKL-40 and other Aplysia egg lectins. Thirty-five amino acids of the N-terminal sequence of AKL-40 were indicated by the single-letter amino acid code. Identical amino acids in the N-terminal sequences of 32–30 kDa Aplysia egg lectins isolated as AKL-a to -d [25] and ADEL [23] were shown as bold letters. X: unidentified, Gray box: skipped.

**Figure 3 marinedrugs-19-00394-f003:**
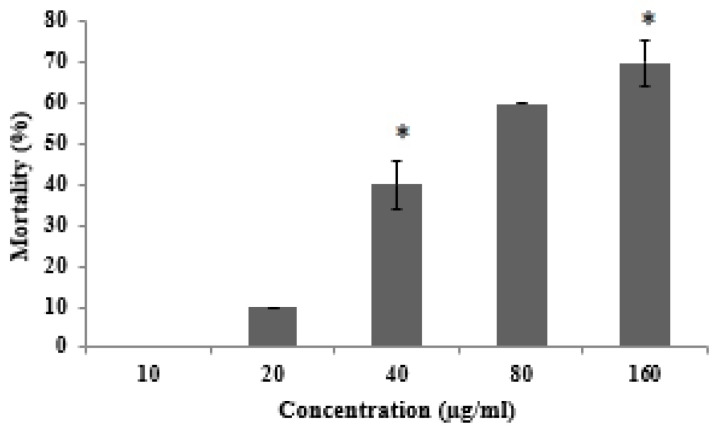
Toxicity of AKL-40 against shrimp nauplii at different concentrations. Data are expressed in mean ± SD (*n* = 10). The results were statistically significant (* *p* < 0.05, when mortality percentage of lectin-treated shrimps were compared to untreated shrimps).

**Figure 4 marinedrugs-19-00394-f004:**
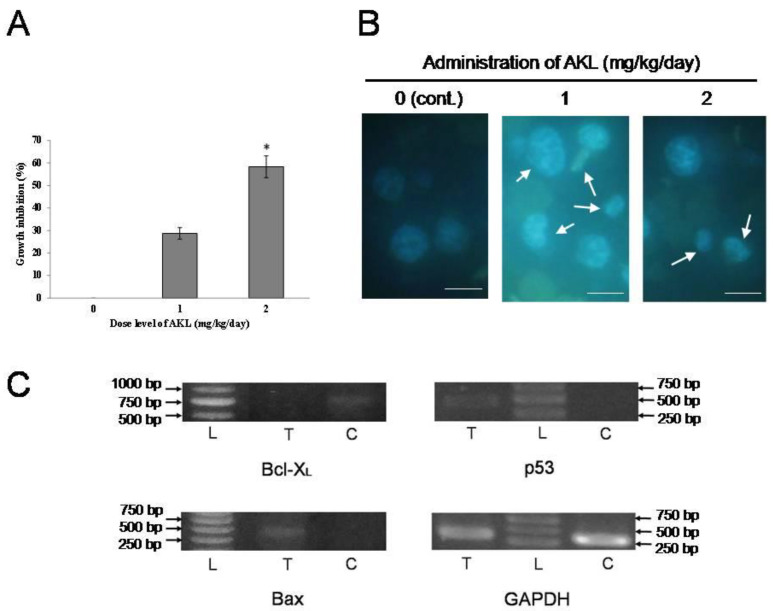
In vivo anticancer activity of AKL-40. (**A**). Growth inhibition in AKL-40 treated mice compared to the untreated mice. The lectin was administrated to mice at doses 0 (untreated), 1 and 2 mg/kg/day (treated). Data are expressed in mean ± SD (*n* = 6). The results were statistically significant (* *p* < 0.05, when cells from lectin-treated mice were compared to cells from untreated mice). (**B**). Change of morphology in Hoechst-stained EAC cells harvested from AKL-40 treated mice. EAC cells from untreated and treated (with 1 mg/kg/day and 2 mg/kg/day of AKL-40, for five days) mice were observed by a fluorescence microscope. White arrows show changes in the morphology of cells. Scale bar: 25 µm. (**C**). Expression of apoptosis-related genes (Bcl-X_L_, p53 and Bax) and GAPDH. L, 1000 bp DNA ladder; T, RNA of EAC cells from AKL-40 treated mice; C, RNA of EAC cells from untreated (control) mice.

**Figure 5 marinedrugs-19-00394-f005:**
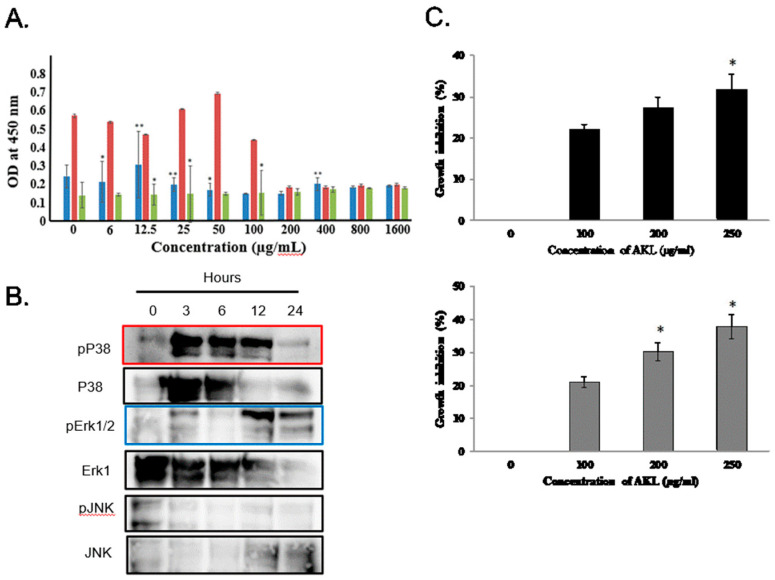
In vitro antiproliferative activity of AKL-40 against different cancer cell lines. (**A**). Antiproliferative activity of AKL-40 against erythroleukemia (K562), human B-lymphoma (Raji), and rat basophilic leukemia (RBL-1) cells. Orange, blue, and gray bars denote the growth of K562, Raji, and RBL-1 cells, respectively, at different concentrations of AKL-40. Data are expressed in mean ± SD. (**B**). Activation of MAPK pathway in AKL-40 treated K562 cells was observed during a period of 24 h. Phosphorylation of P38 was observed from a 3 to 12 h treatment and it diminished at 24 h (surrounded in red). Phosphorylation of Erk1/2 was also shown after the treatment for 12 to 24 h (surrounded in blue). No phosphorylation of JNK was observed. (**C**). Anticancer activity of AKL-40 against EAC and U937 cells. After treating with 100–250 µg/mL of AKL-40, percentages of growth inhibition were determined by MTT assay (*n* = 3, mean ± SD). Gray and black bars indicate EAC and U937 cells, respectively. The results were statistically significant (* *p* < 0.05, when lectin-treated cells were compared to untreated cells). The results were statistically significant (** *p* < 0.01 and * *p* < 0.05, when lectin-treated cells were compared to untreated cells).

**Figure 6 marinedrugs-19-00394-f006:**
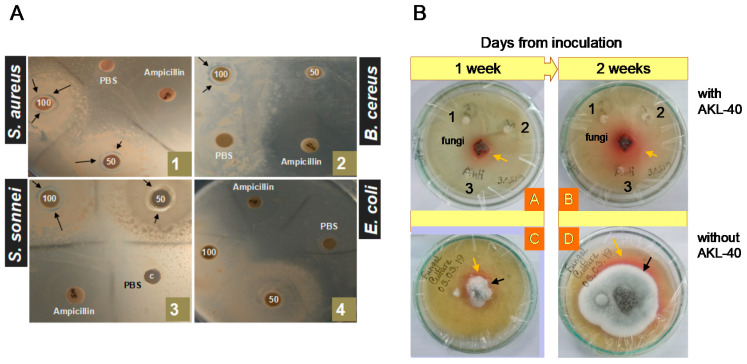
Antimicrobial activity of AKL-40. (**A**). Disc diffusion assay of the antibacterial activity of AKL-40; 1, 2, 3, and 4 shows the activity for *Staphylococcus aureus*, *Bacillus cereus*, *Shigella sonnei*, and *Escherichia coli**,* respectively. Moreover, 50 and 100 on discs soaked with 50 and 100 µg/disc of AKL, respectively. PBS is the negative control (soaked with PBS) and Ampicillin indicated 15 µg of ampicillin (positive control). Black arrows mark the zone of inhibition. (**B**). AKL-40 showed time-dependent antifungal activity against *Talaromyces verruculosus*. A and B show the petri dishes containing three discs soaked with 400 µg/mL of AKL-40 after one week and two weeks of fungal inoculation, respectively). C and D are the petri dishes with no AKL-40 (control) after one week and two weeks of inoculation, respectively. Orange and black arrows indicate the pigment produced by the fungus and its subsequent growth, respectively.

**Table 1 marinedrugs-19-00394-t001:** Primer constructions for apoptosis related and housekeeping genes.

Primer	Forward	Reverse
p53	5′-GCGTCTTAGAGACAGTTGCCT-3′	5′-GGATAGGTCGGCGGTTCATGC-3′
Bax	5′-GGCCCACCAGCTCTGAGCAGA-3′	5′-GCCACGTGGGCGTCCCAAAGT-3′
Bcl-X_L_	5′-TTGGACAATGGACTGGTTGA-3′	5′-GTAGAGTGGATGGTCAGTG-3′
GAPDH	5′-GTGGAAGGACTCATGACCACAG-3′	5′-CTGGTGCTCAGTGTAGCCCAG-3′

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
