# Peer review of "Antiproliferative and Antimicrobial Potentials of a Lectin from Aplysia kurodai (Sea Hare) Eggs"

_marinedrugs, 2021, doi:10.3390/md19070394_

Round 1

Reviewer 1 Report

Dear Editor,

Thanks very much for the opportunity to review the manuscript titled “Antiproliferative and Antimicrobial Potentials of a Lectin from Aplysia kurodai (Sea Hare) Eggs” proposed by Swarna et al.

The manuscript describes the multifaceted activities of AKL-40, a Sea Hare lectin. The text is generally understandable, but revision by a native English-speaking scientist is required. In addition, I find the scope too broad and somewhat shallow. The authors could split this manuscript into two papers: (1) anticancer effects and (2) antimicrobial activities. This would help them provide in-depth analyses to inform the field.

My specific comments are found below:   

Abstract

  1. A sound background should be added at the beginning before the aim.
  2. The average reader would be surprised by AKL-40 at first appearance. Explain what this is.
  3. The description of results seems to be confusing. In vitro and in vivo data are mixed with no clear description.
  4. The section needs language improvement.

Introduction

  1. Lectins are representatives of carbohydrate-binding proteins (representatives of should be deleted).
  2. The opening sentence needs a reference. Other important claims in this section also need references.
  3. "It is believed that galactose residues are present as terminal sugars in lipopolysaccharides" should be reformulated. The structure of LPS is well defined.
  4. "Lectins from mollusks are of considerable interest to researchers in recent years" needs to be reformulated for a proper transition.
  5. "D-galacturonic acid and D-galactose-binding lec-tins with molecular masses 16-34 kDa have been obtained from the same species [10, 19-21] and same genus [22-24]" is rather vague. Which genus? species?
  6. The sentence following the above one is also unclear.
  7. There is no clear hypothesis in this work.
  8. The section needs language improvement.

Results

  1. The authors should better explain their elution method described in Figure 1 (1, 2 and 4 eluted with D-Gal and 3 with D-galacturonate). In Figure 1 I see only one yellow triangle while the legend announces many.
  2. The sections are too short. Mouse data (2.4 to 2.6) should be combined in one paragraph, as well as the associated figures.
  3. Similarly, 2.7 and 2.8 should be combined. The authors should also provide mechanistic data for 2.8, unless there is a rationale for not assessing such signaling in U937 and Ehrlich ascites carcinoma cells.
  4. To avoid very short paragraphs, antibacterial and antifungal activities can also be combined into "antimicrobial activities".

Discussion

  1. Not sure what "It might be interesting to find out the localization of Aplysia egg lectins during each developmental stage of the embryo, suggesting the function of each molecular species of lectins" means.
  2. Would glycosylation change activity? In which direction? The authors should search literature to address this issue. Personally, I think this is not the main point of this manuscript. The authors could focus on AKL-40 activities and compare them with the activities of lectins from other marine organisms.
  3. They should also clearly present the implications of their findings.
  4. This section should be rewritten. A few tips:
  5. Avoid abundant background, which should be moved to the Introduction.
  6. Start the section with a brief description of study aim and main findings.
  7. Briefly describe your findings in light of previous reports.
  8. Provide study limitations.
  9. End with a concise conclusion.

Materials and Methods

  1. Manufacturer description is not uniform: sometimes, city and/or state is missing
  2. Section 4.1. "containing a protease inhibitor": a cocktail has many components. "After repeated centrifugation of the supernatant": at the same speed and for the same time?
  3. Section 4.4. What is under a light regime?
  4. Section 4.5. gm should be g. How many mice were used here?
  5. Section 4.6. Groups should be renamed for clarity. "Weight of the mice in each group was recorded to check the rate of tumor 373
  6. growth": is there any formula for this? '... as described in section 4.6": you likely meant 4.5.
  7. Section 4.7. What was the goal of this assay?
  8. Section 4.9. Description is suboptimal (please refer to other manuscripts to improve your description)
  9. Section 4.12. This method is more for screening. MIC determination is usualy more convincing. To the least, this should be added as a limitation.
  10. Section 4.14. Rewrite this part. "by one way ANOVA using Dunnett ‘t’ test" is unclear.

Author Response

[Abstract] 

  1. A sound background should be added at the beginning before the aim. 

Response: We added a background and wrote down the aim (investigation of biological activities of AKL-40) in the abstract. 

  1. The average reader would be surprised by AKL-40 at first appearance. Explain what this is. 

Response: We mentioned lectins and before introducing AKL-40 wrote about its previously isolated isotypes from sea hare eggs.  

  1. The description of the results seems to be confusing. In vitro and in vivo data are mixed with no clear description. 

Response: We modified the whole abstract to make it clear. 

  1. The section needs language improvement. 

Response: We tried to improve the language of abstract. 

[Introduction] 

  1. Lectins are representatives of carbohydrate-binding proteins (representatives of should be deleted). 

Response: We deleted the phrase. 

  1. The opening sentence needs a reference. Other important claims in this section also need references. 

Response: We provided the necessary references including a new one [6].  

  1. "It is believed that galactose residues are present as terminal sugars in lipopolysaccharides" should be reformulated. The structure of LPS is well defined. 

Response: The sentence is now changed to “Galactose residues present as terminal sugars in lipopolysaccharides (LPS) of bacteria impact the intracellular composition of bacteria and maintain the synthesis of UDP-galactose for LPS [16, 17].” 

  1. "Lectins from mollusks are of considerable interest to researchers in recent years" needs to be reformulated for a proper transition. 

Response: We deleted the sentence. 

  1. "D-galacturonic acid and D-galactose-binding lectins with molecular masses 16-34 kDa have been obtained from the same species [10, 19-21] and same genus [22-24]" is rather vague. Which genus? species? 

Response: We corrected the sentence as “D-galacturonic acid and D-galactose-binding lectins with molecular masses 16-34 kDa have been obtained from the eggs of Aplysia kurodai [10, 19-21] and other Aplysia species (A. depilansA. dactylomela, and A. californica) [22-24].” 

  1. The sentence following the above one is also unclear. 

Response: We modified the sentence as “Primary structures of all these previously purified egg lectins possess a novel triple tandem repeating sequence consisting of 210-230 amino acids and show striking similarities to domain DUF3011 of some uncharacterized bacterial proteins [25].” 

  1. There is no clear hypothesis in this work. 

Response: We clearly mentioned the hypothesis and aims of this work in the newly written last paragraph of the introduction. 

  1. The section needs language improvement. 

Response: We tried to improve the language through re-writing the weakly structured sentences. 

[Results] 

  1. The authors should better explain their elution method described in Figure 1 (1, 2, and 4 eluted with D-Gal and 3 with D-galacturonate). In Figure 1 I see only one yellow triangle while the legend announces many. 

Response: The elution method is described in the ‘Materials and Methods section. We corrected the phrases including ‘yellow (and black) triangles both in the text and legend. 

  1. The sections are too short. Mouse data (2.4 to 2.6) should be combined in one paragraph, as well as the associated figures. 

Response: We combined those sections in one paragraph and modified the figures. We also changed the figure numbers and modified the whole manuscript accordingly. 

  1. Similarly, 2.7 and 2.8 should be combined. The authors should also provide mechanistic data for 2.8, unless there is a rationale for not assessing such signaling in U937 and Ehrlich ascites carcinoma cells. 

Response: We combined sections 2.7 and 2.8. Unfortunately, we ran out of AKL to perform the signaling experiments using U937 and EAC. Purification of AKL was done in Japan as the source (Japanese Sea hare eggs) is available only in Japan. We could not manage to get it in the pandemic situation.   

  1. To avoid very short paragraphs, antibacterial and antifungal activities can also be combined into "antimicrobial activities". 

Response: Sections for antibacterial and antifungal activities have been combined into "antimicrobial activities". 

[Discussion] 

  1. Not sure what "It might be interesting to find out the localization of Aplysia egg lectins during each developmental stage of the embryo, suggesting the function of each molecular species of lectins" means. 

Response: We changed the sentence to “It might be interesting to find out the function of each molecular species of Aplysia egg lectins during each developmental stage of the embryo.” 

  1. Would glycosylation change activity? In which direction? The authors should search literature to address this issue. Personally, I think this is not the main point of this manuscript. The authors could focus on AKL-40 activities and compare them with the activities of lectins from other marine organisms. 

Response: We agree with this comment and deleted the text about glycosylation change activity. We have modified the beginning paragraph of the discussion. In the next paragraphs, we focused on the biological activities of AKL-40 and compared them to other marine lectins. 

  1. They should also clearly present the implications of their findings. 

Response: Possible implications of our findings are now mentioned in the conclusion. 

  1. This section should be rewritten. A few tips: 
  1. Avoid abundant background, which should be moved to the Introduction. 
  1. Start the section with a brief description of study aim and main findings. 
  1. Briefly describe your findings in light of previous reports. 
  1. Provide study limitations. 
  1. End with a concise conclusion. 

Response: We started the discussion with study aims, reduced the background, added more lines to our findings on anticancer and antibacterial activities, provided study limitations in the conclusion and wrote a new conclusion after deleting the previous one.   

[Materials and Methods] 

  1. Manufacturer description is not uniform: sometimes, city and/or state is missing 

Response: The manufacturer description is updated now. For example, NA-18HS rotor (TOMY Co. Ltd., Tokyo, Japan); fraction collector (FRC-10A, Shimadzu Corporation, Tokyo, Japan). 

  1. Section 4.1. "containing a protease inhibitor": a cocktail has many components. "After repeated centrifugation of the supernatant": at the same speed and for the same time? 

Response: We used the ready-to-use concentrated stock solution of protease inhibitors for addition to samples and simply mentioned the company name. We are not sure whether it was a ‘cocktail’ or not. And, we added ‘at the same speed for same duration’ to that sentence. 

  1. Section 4.4. What is under a light regime? 

Response: We corrected the sentence as “The experiment was repeated thrice at 30 °C for 24 h with 6 h of light exposure”. 

  1. Section 4.5. gm should be g. How many mice were used here? 

Response: We changed ‘gm’ to ‘g’. We used eighteen mice in total, six mice in three groups. 

  1. Section 4.6. Groups should be renamed for clarity. "Weight of the mice in each group was recorded to check the rate of tumor 373 

Response: We renamed the groups as ‘A’ or control, ‘B’ or lectin-treated with the lower dose (1 mg/kg/day)’ and ‘C’ or lectin-treated with higher dose (2 mg/kg/day) and corrected the ‘Method’ and ‘Results’ sections accordingly.  

  1. growth": is there any formula for this? '... as described in section 4.6": you likely meant 4.5. 

Response: There is no formula. We take the reading and prepare a chart. The mouse gains weight each day and following the record of their weight, we can ensure successful tumor cell propagation. By the 6th day, we can see visible changes in mice, they show abdominal swelling and a rapid increase in weight. And yes, we corrected ‘section 4.6’ to ‘section 4.5’.    

  1. Section 4.7. What was the goal of this assay? 

Response: By this assay, we try to look for morphological changes in the lectin-treated EAC cells. If we observe signs of apoptosis-like nuclear blebbing, cell shrinkage or formation of apoptotic bodies; we go for the next experiment (expression of apoptotic genes).    

  1. Section 4.9. Description is suboptimal (please refer to other manuscripts to improve your description) 

Response: We modified the description and described the cytotoxic activity experiment. 

  1. Section 4.12. This method is more for screening. MIC determination is usually more convincing. To the least, this should be added as a limitation. 

Response: We added this limitation in the conclusion. 

  1. Section 4.14. Rewrite this part. "by one-way ANOVA using Dunnett ‘t’ test" is unclear. 

Response: This part is re-written as “To test differences between experimental conditions, one-way ANOVA and Dunnett’s post-test correction was used.” 

Reviewer 2 Report

The manuscript “Antiproliferative and Antimicrobial Potentials of a Lectin from Aplysia kurodai (Sea Hare) Eggs” presents a relatively straightforward study characterizing several aspects of biological activity of the 40 kDa lectin. The study has used multiple methods, however the clarity of the report has issues at several places which affects my enthusiasm:

- Lines 101-103: Where did this statement about glycosylation come from?- “This result suggested that AKL-40 was a unique polypeptide whereas different post-translational modifications like glycosylation took place in case of AKL-a to -d”.

- Many Figure legends: Ambiguous statement that “The results were statistically significant (p < 0.05).” Post hoc stats details are required.

- Line 145: wrong title; this subsection presents PCR results. The quality of the results is very poor, larger part of gels are required; weird highlight of transcript; organize gels uniformly.

- Figure 7: what is plotted on Y-axis? Strange placement of cell line names on the figure; the absence of loading control on western blots?

- Line 172: A terminology issue – do not call cancer cell agglutination as hemagglutination.

- Table 1: specify the positions of PCR primer sequences in corresponding mRNAs and mRNAs accession numbers.

- Lines 416-423: the description of this method is unclear. Did you mix human erythrocytes with EAC and U937 cells? It would make no sense for me.

- Lines 424-435: the description of this method is unclear because both U937 and EAC cells are growing is suspension. How did you combine the step “the aliquot was carefully removed from each well.” with subsequent steps? How did you control cell concentration?

- Origin of all cell lines should be specified (ATCC? Or others?).

- General comment to the manuscript: the authors claim they study the lectin and its activity, however no carbohydrate inhibitors were used to confirm that any of the observed effects were glycan-specific. This is an important weakness of this study which should be addressed.

Author Response

  1. Lines 101-103: Where did this statement aboutglycosylation come from?- “This result suggested that AKL-40 was a unique polypeptide whereas different post-translational modifications like glycosylation took place in case of AKL-a to -d”. 

Response: It was just speculation. So, we deleted the statement.  

  1. Many Figure legends: Ambiguous statement that “The results were statistically significant (p < 0.05).” Post hoc stats details are required.

Response: We performed Dunnett’s test (which is a post hoc analysis) for Figs. 4, 7, and 8 as we had fixed control groups to compare to all other groups. Fig. 3 was done by Finney’s method which also had a control group. We tried for the Bonferroni correction but unfortunately do not have enough expertise for that statistical analysis.  

  1. Line 145: wrong title; this subsection presents PCR results. The quality of the results is very poor, larger part of gels are required; weird highlight of transcript; organize gels uniformly. 

Response: The title is corrected as “Expression of apoptosis-related genes from AKL-40 treated EAC cells”. We changed the figure (Fig. 6) and removed the highlight of transcripts.  

  1. Figure 7: what is plotted on Y-axis?Strange placement of cell line names on the figure; the absence of loading control on western blots? 

Response: ‘Absorbance (OD) at 450 nm’ is plotted on Y-axis. It was determined by a WST assay. We modified the method section describing the method clearly. Figure 7 is now replaced with a new one showing more genes and the loading control. 

  1. Line 172: A terminology issue – do not call cancer cell agglutination as hemagglutination.

Response: We corrected it as ‘minimum agglutination concentration of EAC and U937 cells by AKL-40’. 

  1. Table 1: specify the positions of PCR primer sequences in corresponding mRNAs and mRNAs accession numbers.

Response: We don’t have this information as we didn’t design the primers. We bought the primers from Integrated DNA Technologies (IDT), Singapore. We added this information in the manuscript. Here are the research articles we followed to order the primers: 

  1. https://journals.plos.org/plosone/article?id=10.1371/journal.pone.0167536
  2. https://www.hindawi.com/journals/ecam/2020/9145626/tab1/. 
  3. Lines 416-423: the description of this method is unclear. Did you mix human erythrocytes with EAC and U937 cells? It would make no sense for me.

Response: The method is now correctly described. 2% suspensions of EAC and U937 cells were used, not erythrocytes. 

  1. Lines 424-435: the description of this method is unclear because both U937 and EAC cells are growing in suspension. How did you combine the step “the aliquot was carefully removed from each well.” with subsequent steps? How did you control cell concentration?

Response: We add 100 µl of EAC and U937 cells to each well (5×105cells/well). After a 24-hour incubation period, we take out the titer plates without disturbing those and can find a cell mass lying to the surface of each well. Yes, they grow in suspension but it is possible to get those settled down (we can see the mass) and remove the aliquot from wells. At the beginning of the next steps and before taking the reading by a titer plate reader, we gently shake the wells and get those cells back in the suspension form. 

  1. Origin of all cell lines should be specified (ATCC? Or others?).

Response: Ehrlich ascites carcinoma cell line was propagated and maintained in the Animal Laboratory, Department of Biochemistry and Molecular Biology, University of Rajshahi, Bangladesh. U937 cell line was collected from Yokohama City University, Japan (ATCC CRL-3253). This information is now included in the Methods section.  

  1. General comment to the manuscript: the authors claim they study the lectin and its activity, however, no carbohydrate inhibitors were used to confirm that any of the observed effects were glycan-specific. This is an important weakness of this study that should be addressed. 

Response: In our previous article (Molecules, 2014), this particular lectin showed inhibition of antimicrobial activity in the presence of galactose sugar in the culture medium (doi:10.3390/molecules190913990). That’s why we didn’t focus on it this time. We are mentioning this limitation in the conclusion. 

Reviewer 3 Report

The language needs some polishing.

N=?in fig 3.

Do the inhibiting sugars D-galacturonic acid and D-galactose. abolish to any extent  the antiproliferative activity and antimicrobial activity of the lectin?

Does the lectin affect viability and proliferation of normal cells?

How does the lectin compare with other Sea Hare lectins in yield,specific activity and other aspects?

Please further elaborate on the novelty of the findings in the light of the existing literature.

Author Response

  1. The language needs some polishing.

Response: We tried to improve the language. 

  1. N=?in fig 3. 

Response: We took ten brine shrimp nauplii in each tube which is mentioned in the method. So, N = 10. It is now included in the figure 3 legend.  

  1. Do the inhibiting sugars D-galacturonic acid and D-galactose abolish to any extent the antiproliferative activity and antimicrobial activity of the lectin? 

Response: In our previous article (Molecules, 2014), this particular lectin showed inhibition of antimicrobial activity in the presence of galactose sugar in the culture medium (doi:10.3390/molecules190913990). That’s why we didn’t focus on it this time. We are mentioning this limitation in the conclusion. However, cancellation effect of the antimicrobial activity by this lectin was once visible in this study but unfortunately we lost the pictures 

  1. Does the lectin affect viability and proliferation of normal cells?

Response: Though lectins usually do not affect the viability and proliferation of normal cells (Fik E, Wołuń-Cholewa M, Kistowska M, Warchoł JB, Goździcka-Józefiak A. Effect of lectin from Chelidonium majus L. on normal and cancer cells in culture. Folia Histochem Cytobiol. 2001;39(2):215-6. PMID: 11374832), it would have been better to check this lectin. But working with normal cells demands more facilities than working with cancer cells and due to the lack of facilities, we were unable to check the effects against normal cells. 

  1. How does the lectin compare with other Sea Hare lectins in yield, specific activity, and other aspects? 

Response: Before AKL-40, two other AKLs (AKL and AKL-2) were purified from Aplysia kurodai eggs our lab [i-ii]. AKL, the first lectin of this series had a molecular weight of 56 kDa, specific activity and purification folds were 268 and 172. AKL-2 (16 kDa) had a specific activity of 585 whereas the purification fold was 89. But this lectin, AKL-40 was rather accidentally purified in 2014 [iii]So, we could not focus on its specific activity and purification fold at that time. Even after 6 years, when we realized it, we focused on its biological activities, not on the purification scheme. Now we can find that the specific activity of AKL-40 was around 350 and the purification fold was 120. So, these values are similar to those of AKL, the first one isolated. The molecular weights of these two lectins are also close (56 and 40 kDa). However, we did not show the purification table after so many years of its purification in this manuscript.  

The specific activity and purification fold of Aplysia depilans eggs were 6400 and 125 [iv]ADELanother egg lectin from Aplysia dactylomela (57 kDa) had the following values: Specific activity of 5688 and purification fold of 8 [v]. We could not find the purification table for Aplysia californica egg lectin (34 kDa) [vi] 

Almost all of these egg lectins have a strong affinity for D-galactose and galacturonic acid. And, they are glycoprotein in nature. In the case of AKL-40, its glycoprotein nature was not checked. 

  1. i.Kawsar, M.A.; Matsumoto, R.; Fujii, Y.; Yasumitsu, H.Dogasaki, C.Hosono, M.; Nitta, K.Hamako, J.; Matsui, T.; Kojima, N.; Ozeki, Y. Purification and biochemical characterization of D-galactose binding lectin from Japanese sea hare (Aplysia kurodai) eggs. Biochemistry (Mosc). 2009, 74709-716. doi: 10.1134/s0006297909070025. 
  2. ii.KawsarSMA, Matsumoto R, Fujii Y, Matsuoka H, Masuda N, Chihiro I,Yasumitsu H, Kanaly RA, Sugawara S, Hosono M, Nitta K, Ishizaki N, Dogasaki C, Hamako J, Matsui T, Ozeki  2011. Cytotoxicity and glycan-binding profile of a D-galactose-binding lectin from the eggs of a Japanese sea hare (Aplysia kurodai). Protein J 30:509–519. doi: 10.1007/s10930-011-9356-7.  

iiiHasan, I.; Watanabe, M.; Ishizaki, N.; Sugita-Konishi, Y.; Kawakami, Y.; Suzuki, J.; Dogasaki, C.; Rajia, S.; Kawsar, S.M.A.; Koide, Y.; Kanaly, R.A.; Sugawara, S.; Hosono, M.; Ogawa, Y.; Fujii, Y.; Iriko, H’; Hamako, J.; Matsui, T.; Ozeki, Y. A galactose-binding lectin isolated from Aplysia kurodai (sea hare) eggs inhibits streptolysin-induced hemolysisMolecules. 2014, 19, 13990–14003. doi: 10.3390/molecules190913990. 

  1. iv.Gilboa-Garber, N.;Susswein, A.J.; Mizrahi, L.; Avichezer, D. Purification and characterization of the gonad lectin of Aplysia depilansFEBS Lett. 1985, 181, 267-270. doi: 10.1016/0014-5793(85)80273-8. 
  2. v.Carneiro, R.F.; Torres, R.C.; Chaves, R.P.; deVasconcelos, M.A.; de Sousa, B.L.; Goveia, A.C.; Arruda, F.V.; Matos, M.N.; Matthews-Cascon, H.; Freire, V.N.; Teixeira, E.H.; Nagano, C.S.; Sampaio, A.H. Purification, biochemical characterization, and amino acid sequence of a novel type of lectin from Aplysia dactylomela eggs with antibacterial/antibiofilm p Mar. Biotechnol. (NY). 2017, 19, 49-64. doi: 10.1007/s10126-017-9728-x. 
  3. vi.Wilson, M.P.;Carrow, G.M.; Levitan, I.B. Modulation of growth of Aplysia neurons by an endogenous lectin. J. Neurobiol1992, 23, 739-750doi: 10.1002/neu.480230611. 
  4. Please further elaborate on the novelty of the findings in the light of the existing literature.

Response: So far a number of AKLs (and other Aplysia egg lectins) have been isolated. Previous works were limited to purification and biochemical characterization. In recent works, biological activities like cytotoxicity or antibacterial/antibiofilm properties are studied. One paper showed the primary structures of AKLs and their involvement in the developmental stage of A. kurodai [Motohashi, S.; Jimbo, M.; Naito, T.; Suzuki, T.; Sakai, R.; Kamiya, H. Isolation, amino acid sequences, and plausible functions of the galacturonic acid-binding egg lectin of the Sea hare Aplysia kurodaiMar. Drugs. 2017, 15, 161. doi: 10.3390/md15060161]. In this study, we are re-introducing a novel lectin from Aplysia kurodai eggs (though it was isolated and published in 2014). Its N-terminal sequence was analyzed and compared to the Motohashi et al article. We are reporting the in vivo and in vitro anticancer activity of this lectin against four cell lines for the first time. Antibacterial and antifungal properties of this lectin were also studied. And, we are suggesting that lectins are involved not only in embryo development but also in embryo defense. In the last paragraph of the introduction of the revised version, we have tried to mention the novelty of all these findings clearly. 

Round 2

Reviewer 1 Report

The authors have addressed most of my concerns.

However,  a few items remain:

1. AKL-40 still surprises the reader (to the least it should be fully defined at first appearance)
2. The authors said "We tried to improve the language of abstract": they should find someone well versed in the English language to help them.

3. The authors said; "The elution method is described in the ‘Materials and Methods section". Since the section is after Discussion, the method should be briefly mentioned in Results, especially since it is critical for the understanding of the section

Author Response

1. AKL-40 still surprises the reader (to the least it should be fully defined at first appearance)

Response: To solve this issue, we inserted the following sentence in the abstract: “Similar to its previously purified isotypes with different molecular masses (32/30 and 16 kDa), the 40 kDa Aplysia kurodai egg lectin (or AKL-40) binds to D-galacturonic acid and D-galactose sugars”.

2. The authors said "We tried to improve the language of abstract": they should find someone well versed in the English language to help them.

Response: We took the help of a researcher (well-versed in English language) to brush-up the abstract.

3. The authors said; "The elution method is described in the ‘Materials and Methods section". Since the section is after Discussion, the method should be briefly mentioned in Results, especially since it is critical for the understanding of the section

Response: According to the suggestion of reviewer-2, we deleted Fig. 1 and relevant sentences. However, for better understanding we added the following sentence in this ‘Results’ section: “After applying the crude protein sample, the melibosyl-agarose affinity chromatography column was washed with TBS and bound lectins were eluted with 10 mM D-galacturonic acid or D-galactose-containing TBS.”

Reviewer 2 Report

This manuscript still contains many sloppy or poor places, which discourage my enthusiasm.

1) Figure 1A represent poor SDS-PAGE gels and if authors wants to keep them, I would advice to move them to a supplementary materials. Also, the logic of this figure suggests to show the elution profile of proteins separated by using Sephacryl S-200 gel filtration column. Also, the authors should be consistent with presentation of Figure 1B gels using blue or grey colours.

2) Figure 3 and others and Statistical analysis section indicate that the results were statistically significant at p < 0.05. Statements like this have no sense. Make sure to show on the figures and explain in detail significant differences between which values are considered.

3) section 2.4. : needs revision for clarity since the authors are talking here about ‘lectin-treated cells’, while you treated mice with this lectin.

4) Figure 4: very poor PCR gels. They need to be re-done. Also, the authors did not fully consider my previous comments about PCR primers. The sequences should be verified by BLAST, even if they were received from IDT, and the positions of oligoprimers should be reported in the Table1 together with names of reference mRNA sequences of the related genes. Also report the size of PCR amplicons in b not kb. Also, explain why PCR amplicons were kept at 20oC.

5) Figure 5: label the western blots with an individual letter.

6) line 332: specify the catalog number of ‘melibiosyl-agarose affinity column’.

7) section 4.5 and 4.6: combine these sections together and must refer to the animal use protocol approved by your organization. Also, A, B, C animal groups are not specified in the result section.

8) line 425 (section 4.10): very confused statement that the authors used ‘2% suspension solutions of EAC and U937 cells’. Most likely it has no sense in this case and need very clear explanations. How was 2% cell suspension was made? All details must be described.

9) Section 4.11: again, this is very confused description of working process with cells in suspension. What are ‘aliquot’ is absolutely not clear. All details of this method must be described the way that everybody can reproduce. All volumes must be specified.

10) General comment: all figures should edited making sure to maintain consistency of fonts and sizes.

11) General comment: authors ignore the multitarget effects of galactose-binding lectins including galectins, which can be produced by cancer cells (doi: 10.1007/s00018-015-2008-x ) and moreover tumor-produced galectins can kill immune T-cells, which is one of the mechanism how cancer cells manage in avoiding immune surveillance ( DOI: 10.1007/s10719-012-9379-0 ). These options should be at least mention either in the introduction and discussion.

Author Response

1) Figure 1A represent poor SDS-PAGE gels and if authors wants to keep them, I would advice to move them to a supplementary materials. Also, the logic of this figure suggests to show the elution profile of proteins separated by using Sephacryl S-200 gel filtration column. Also, the authors should be consistent with presentation of Figure 1B gels using blue or grey colours.

Response: We deleted Figure 1A according to the suggestion. We also deleted the related information of elution profile of those proteins from results and discussion. Figure 1B is now renamed as Figure 1 and gels are now presented using gray colours.

2) Figure 3 and others and Statistical analysis section indicate that the results were statistically significant at p < 0.05. Statements like this have no sense. Make sure to show on the figures and explain in detail significant differences between which values are considered.

Response: We added significant differences between considered values in the figures and their legends.

3) section 2.4. : needs revision for clarity since the authors are talking here about ‘lectin-treated cells’, while you treated mice with this lectin.

Response: We revised the section and replaced ‘lectin-treated cells’ with ‘cells from lectin-treated/AKL-40 treated mice’. We did the same for ‘cells from untreated/control mice’.

4) Figure 4: very poor PCR gels. They need to be re-done. Also, the authors did not fully consider my previous comments about PCR primers. The sequences should be verified by BLAST, even if they were received from IDT, and the positions of oligoprimers should be reported in the Table1 together with names of reference mRNA sequences of the related genes. Also report the size of PCR amplicons in b not kb. Also, explain why PCR amplicons were kept at 20oC.

Response: Unfortunately, we ran out of AKL to perform the signaling experiments by PCR. Purification of AKL was done in Japan as the source (Japanese Sea hare eggs) is available only in Japan. We could not manage to get it in the pandemic situation.

We tried to verify the sequences by BLAST. They matched 100% with the database (please see the attached file) but we could not find the exact sequence ID due to our lack of expertise in bioinformatics. That is why we could not insert the names of reference mRNA of related genes in Table 1. Hope you will consider it.

We are now reporting the size of PCR amplicons in bp, instead of kb.

PCR amplicons were kept in room temperature (20oC). It is obviously true that keeping those at lower temperature, improves their stability a lot. But, it also puts strain on the thermal block (https://www.coleparmer.com/blog/2019/07/30/is-a-4c-holding-temperature-for-a-pcr-reaction-necessary/). So, we keep it at 20 oC to save the life of thermal cycler and if we need to store the completed reactions, we preserve those at -20oC.

5) Figure 5: label the western blots with an individual letter.

Response: The western blots were labeled with a ‘C’.

6) line 332: specify the catalog number of ‘melibiosyl-agarose affinity column’.

Response: We corrected the company name and specified the catalog number of the column as “Catalog No. J707, Lot No. 30302C, J-Oil Mills Inc, Tokyo, Japan”

7) section 4.5 and 4.6: combine these sections together and must refer to the animal use protocol approved by your organization. Also, A, B, C animal groups are not specified in the result section.

Response: We combined these two sections and referred the approval of animal use protocol. We specified group A, B and C in the result (section 2.4) of the revised version. However, we marked it with red color again.

8) line 425 (section 4.10): very confused statement that the authors used ‘2% suspension solutions of EAC and U937 cells’. Most likely it has no sense in this case and need very clear explanations. How was 2% cell suspension was made? All details must be described.

Response: We removed the incorrect information and modified the section. Cells were counted using a hemocytometer and around 5×105 EAC and U937 cells in RPMI-1640 media were seeded in each well.

9) Section 4.11: again, this is very confused description of working process with cells in suspension. What are ‘aliquot’ is absolutely not clear. All details of this method must be described the way that everybody can reproduce. All volumes must be specified.

Response: We replaced the confusing word ‘aliquot’ by ‘clear supernatant’ to make it clear.

10) General comment: all figures should edited making sure to maintain consistency of fonts and sizes.

Response: We edited all figures maintaining consistency of fonts and sizes. As we combined too many figures (with subsections) according to the instruction of reviewers, sizes became inconsistent.

11) General comment: authors ignore the multitarget effects of galactose-binding lectins including galectins, which can be produced by cancer cells (doi: 10.1007/s00018-015-2008-x ) and moreover tumor-produced galectins can kill immune T-cells, which is one of the mechanism how cancer cells manage in avoiding immune surveillance ( DOI: 10.1007/s10719-012-9379-0 ). These options should be at least mention either in the introduction and discussion.

Response: We mentioned the multitarget effects of lectins by cancer cells in the discussion and added the above mentioned references.

Round 3

Reviewer 2 Report

Authors have tried to address or answer my questions and I have no other comments.